# Self-supervised learning of Maximum Manifold Capacity Representations

## Abstract

Self-supervised Learning (SSL) provides a strategy for constructing useful representations of images without relying on hand-assigned labels. Many such methods aim to learn a function that maps distinct views of the same scene or object to nearby points in the representation space. These methods are often justified by showing that they optimize an objective that is an approximation of (or correlated with) the mutual information between representations of different views. Here, we recast the problem from the perspective of *manifold capacity*, a measure that has been used to evaluate the classification capabilities of a representation. Specifically, we develop a contrastive learning framework that aims to maximize the number of linearly separable object manifolds, yielding a Maximum Manifold Capacity Representation (MMCR). We apply this method to unlabeled images, each augmented by a set of basic transformations, and find that it learns meaningful features using the standard linear evaluation protocol. Specifically, we find that MMCRs support performance on object recognition comparable or better than recently developed SSL frameworks, while providing more robustness to adversarial attacks. Finally, empirical analysis reveals the means by which compression of object manifolds gives rise to class separability.

## 1 Introduction

Natural images lie, at least locally, within manifolds whose intrinsic dimensionality is low relative to that of their embedding space (the set of pixel intensities). Nevertheless, these manifolds are enormously complex, as evidenced by the variety of natural scenes. A fundamental goal of machine learning is to extract these structures from observations, and use them to perform inference tasks. In the context of recognition, consider the *object submanifold*, $\mathcal{M}_j$, which consists of all images of object $j$ (for example, those taken from different camera locations, or under different lighting conditions). Object recognition networks act to map images within a submanifold to nearby representations, relative to images from other submanifolds, and this concept has been effictively exploited in recent self-supervised learning (SSL) methods (Zbontar et al., 2021; Chen et al., 2020; Caron et al., 2020; Bachman et al., 2019; Wang & Isola, 2020; Wang et al., 2022). Most of these operate by minimizing pairwise distances between images within submanifolds, while contrastively maximizing pairwise distances between images in different submanifolds.

A parallel effort in computational neuroscience has aimed to characterize manifolds in neural representations, and their relationship to underlying neural circuits (Kriegeskorte & Kievit, 2013; Chung & Abbott, 2021). Studies in various modalities have identified geometric structures in neural data that are associated with behavioral tasks (Bernardi et al., 2020; DiCarlo & Cox, 2007; Hénaff et al., 2021; Gallego et al., 2017; Nieh et al., 2021), and explored metrics for quantifying these representation geometries.

Here, we make use of a recently developed measure of *manifold capacity*, rooted in statistical physics (Chung et al., 2018), which has been used to evaluate how many manifolds can be linearly separated within the representation space of various models. We develop a simplified form of this meausre, and incorporate it into a novel contrastive objective, that maximizes the extent of the global image manifold while minimizing that of constituent object manifolds. We apply this to an unlabeled set of images, each augmented to form a small set of samples from their corresponding manifold. We show that the learned representations

- support high-quality object recognition, when evaluated using the standard linear evaluation (Chen et al., 2020) paradigm (i.e., training a linear classifer to operate on the output of the unsupervised network). In particular, performance is approximately matched to that of other recently proposed SSL methods.
- extract semantically relevant features from the data, that can be revealed by examining the learning signal derived from the unsupervised task
- have interpretable geometric properties
- are more robust to adversarial attack than those of other recently proposed SSL methods.

## 1.1 RELATED WORK

Our methodology is closely related to and inspired by recent advances in contrastive self-supervised representation learning (SSL), but has a distinctly different motivation. Many recent frameworks craft objectives that are designed to maximize the mutual information between representations of different views of the same object (Oord et al., 2018; Chen et al., 2020; Oord et al., 2018; Tian et al., 2020; Bachman et al., 2019)). However, estimating mutual information in high dimensional feature spaces (which is the regime of modern deep learning models models) has been difficult to compute historically (Belghazi et al., 2018), and furthermore it is not clear that more closely approximating mutual information in the objective produces improved representations (Wang & Isola, 2020). [1] By contrast, capacity estimation theories operate in the regime of large ambient dimension as they are derived in the "large N (thermodynamic) limit" (Chung et al., 2018; Bahri et al., 2020). Therefore we test whether one such measure, which until now had been used to evaluate the quality of representations, might be useful as objective function in SSL.

Operationally, many existing methods are optimized to minimize some notion of distance between the representations of different augmented views of the same image, while maximizing the distance between representations of (augmented views of) distinct images (these are thought of as encouraging alignment and uniformity in the framework of Wang & Isola (2020)). When taking the view that different views of an image form a continuous manifold that we aim to compress, the distance between two randomly sampled points from said manifold seems a strange choice for the size metric to optimize for. Perhaps unsurprisingly it has been demonstrated on multiple occasions, notably by the success of the "multi-crop," strategy implemented SwAV (Caron et al., 2020) and earlier in the contrastive multiview coding work by Tian et al. (2020)). However most commonly the use of multiple views is such that the objective effectively becomes a Monte Carlo estimate with more than one sample of the same pairwise distance function.

Rather than using the mean distance or cosine similarity between pairs of points, we use a nuclear norm as a combined measure of size and dimensionality of groups of points, an idea that is strongly motivated by learning theory. The nuclear norm has been previously used to induce or infer low rank structure in the representation of data, for example, in Hénaff et al. (2015); Wang et al. (2022); Lezama et al. (2018). In particular, Wang et al. (2022) employ the nuclear norm as a regularizer to supplement an InfoNCE loss. Our approach represents a more radical departure from the traditional InfoNCE loss, as we will detail below. Rather than pair a low-rank prior with a logistic regression-based likelihood, we make the more symmetric choice of employing a *high rank* likelihood. This allows the objective to explicitly discourage dimensional collapse, a well known issue in SSL (Jing et al., 2021).

Another consequence of encouraging maximal rank over the dataset is that the objective encourages the representation to form a simplex equiangular tight frame (sETF). sETFs have been shown to be optimal in terms of cross-entropy loss when features lie on the unit hypersphere (Lu & Steinerberger, 2020), and such representations can be obtained in the supervised setting when optimizing either the traditional cross-entropy loss or a supervised contrastive loss (Papyan et al., 2020; Graf et al., 2021). Recent work has shown that many popular objectives in SSL can be understood as different methods of approximating a loss function whose minima form sETFs (Dubois et al., 2022). Our approach is novel, in that it encourages sETF representations by directly optimizing the distribution of singular values, rather than minimizing a cross-entropy loss.

---

[1] Barlow Twins (Zbontar et al., 2021) notably avoids the curse of dimensionality because their objective effectively estimates information under a Gaussian parameterization rather than doing so non-parametrically as in the InfoNCE loss. Our method also makes use of Guasian/second order parameterizations, as detailed below.

## 2 MAXIMUM MANIFOLD CAPACITY REPRESENTATIONS

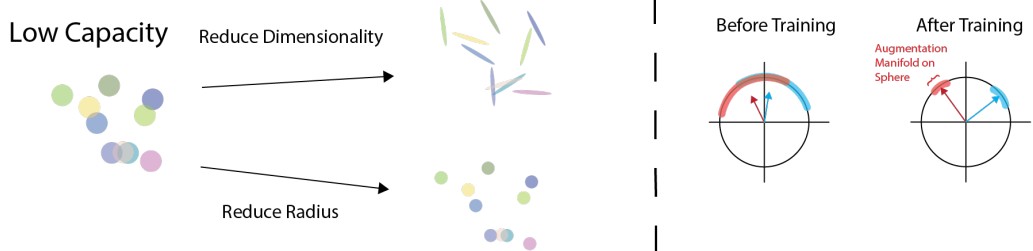

Figure 1: Two dimensional illustrations of high and low capacity representations. Left: the capacity (linear separability) of a random set of elliptical regions can be improved, either by reducing their sizes (while maintaining their dimensionalities), or by reducing their dimensionalities (while maintaining their sizes). Right: the objective proposed in this paper aims to minimize the nuclear norm (product of size and sqrt dimensionality) of normalized data vectors (ie., lying on the unit sphere). Before training the manifolds have a large extent and thus the matrix of their corresponding centroid vectors has low nuclear norm. After training the capacity is increased. The manifolds are compressed and repelled from each other, resulting in centroid matrix with larger nuclear norm and lower similarity.

### 2.1 MANIFOLD CAPACITY THEORY

Consider a set of $P$ manifolds embedded in a feature space of dimensionality $D$, each assigned a random binary class label. Manifold capacity theory is concerned with the question: what is the largest value of $\frac{P}{D}$ such that there exists (with high probability) a hyperplane separating the two classes? Recent theoretical work has demonstrated that there exists a critical value, dubbed the manifold capacity $\alpha_C$, such that when $\frac{P}{D} < \alpha_C$ the probability of finding a separating hyperplane is approximately $1.0$, and when $\frac{P}{D} > \alpha_C$ the probability is approximately $0.0$. Furthermore, $\alpha_C$ can be accurately predicted from three key quantities: (1) the manifold radius $R_M$, which measures the size of the manifold relative to its distance from the origin, (2) the manifold dimensionality $D_M$ which estimates the number of dimensions along which a manifold has significant extent, and (3) the centroid correlation (if the positions of manifolds are correlated with each other they will be more difficult to separate). In particular, when the centroid correlation is low the manifold capacity can be approximated by $\phi(R_M \sqrt{D_M})$ where $\phi(\cdot)$ is a monotonically decreasing function.

For manifolds of arbitrary geometry calculating the manifold radii and dimensionalities involves an iterative process that alternates between determining the set of "anchor points" on each manifold that are relevant for the classification problem, and computing the statistics of random projections of these anchor points (Cohen et al., 2020). This process is both computationally costly and non-differentiable, and therefore not suitable for use as an objective function. For more detail on the general theory see A.2. However, if the submanifolds are assumed to be elliptical in shape there is an analytical expression for each of these,

$$R_M = \sqrt{\sum_i \lambda_i^2}, \qquad D_M = \frac{(\sum_i \lambda_i)^2}{\sum_i \lambda_i^2}, \tag{1}$$

where the $\lambda_i^2$ are the eigenvalues of the covariance matrix of points on the manifold. For reference, for a batch of 100 128-D manifolds with 100 points sampled from each, computing these elliptical-assuming measures is approximately 500 times faster in terms of wall-clock time.

Using these definitions for manifold radius and dimensionality we can write the capacity as $\alpha_C = \phi(\sum_i \sigma_i)$ where $\sigma_i$ are the *singular values* of a matrix containing points on the manifold (which are the square roots of the eigenvalues of the covariance matrix). In this form, the sum is the $L_1$ norm of the singular values, known as the *Nuclear Norm* of the matrix. When used as an objective function, this measure will prefer sparse solutions (i.e., a small number of non-zero singular values) corresponding to low dimensionality. It is worth comparing this objective to another natural

candidate for quantifying size: the determinant of the covariance matrix. The determinant is equal to the product of the eigenvalues (which captures the squared volume of the corresponding ellipse), but lacks the preference for lower dimensionality that comes with the Nuclear Norm. Specifically, since the determinant is zero whenever one (or more) eigenvalue is zero, it cannot distinguish zero-volume manifolds of different dimensionality. In Yu et al. (2020), lossy coding rate (entropy) is used as a measure of compactness, which simplifies to the log determinant under a Gaussian model Ma et al. (2007). In that work, the identity matrix is added to a multiple of the feature covariance matrix before evaluating the determinant, which solves the dimensionality issue described above.

## 2.2 OPTIMIZING MANIFOLD CAPACITY

Manifold Capacity has been previously used to evaluate and compare network representations (Chung et al., 2018; Cohen et al., 2020). Here, we explore its use as an objective function for self-supervised learning. For each input image (notated as a vector $\mathbf{x}_b \in \mathbb{R}^{\mathbb{D}}$) we generate $K$ samples from the corresponding manifold by applying a set of random augmentations (each drawn from the same distribution), yielding a manifold sample matrix $\tilde{\boldsymbol{X}}_b \in \mathbb{R}^{D \times K}$. Each augmented image is transformed by a Deep Neural Network, which computes nonlinear function $f(\mathbf{x}_b; \theta)$ parameterized by $\theta$, and the $d$-dimensional responses are projected onto the unit sphere yielding manifold response matrix $\boldsymbol{Z}_b \in \mathbb{R}^{d \times K}$. The centroid $\boldsymbol{c}_b$ is approximated by averaging across the columns. For a set of images $\{\mathbf{x}_1, ..., \mathbf{x}_B\}$ we compute normalized response matrices $\{\boldsymbol{Z}_1, ..., \boldsymbol{Z}_B\}$ and assemble their corresponding centroids into matrix $\boldsymbol{C} \in \mathbb{R}^{d \times B}$.

Given the responses and their centroids, the loss function is expressed as:

$$\mathcal{L} = -||\boldsymbol{C}||_* + \lambda \mathbb{E}_b[||\boldsymbol{Z}_b||_*] \tag{2}$$

where $|| \cdot ||_*$ indicates the nuclear norm and $\lambda$ is a tradeoff parameter. The first term maximizes the extent of the "centroid manifold" to encourage separability while the second term encourages object manifold compression .

**Compression by Maximizing Centroid Nuclear Norm Alone** Interestingly, the first term also has a compressive effect. This is because each centroid vector, as a mean of unit vectors, has norm that is linearly related to the average cosine similarity of vectors of said unit vectors. Specifically,

$$||\boldsymbol{c}_b||^2 = \frac{1}{K} + \frac{2}{K^2} \sum_{k=1}^{K} \sum_{l=1}^{k-1} \boldsymbol{z}_{b,k}^T \boldsymbol{z}_{b,l} \tag{3}$$

Here $\boldsymbol{z}_{b,i}$ denotes the representation of the $i^{th}$ augmentation of $\boldsymbol{x}_b$. Then because the nuclear norm is bounded below by the Frobenius norm (Recht et al., 2010), $||\boldsymbol{C}||_* \geq ||\boldsymbol{C}||_F = \sqrt{\sum_{b=1}^{B} ||\boldsymbol{c}_b||^2}$, maximizing the centroid nuclear norm optimizes an upper bound on the norms of centroid vectors, thus encouraging intra-object manifold similarity. We can gain further insight by considering how the distribution of singular vectors of a matrix depends on the norms and pairwise similarities of the constituent column vectors. While no closed form solution exists for the singular values of an arbitrary matrix, the case where the matrix is composed of two column vectors can provide useful intuition. If $\boldsymbol{C} = [\boldsymbol{c}_1, \boldsymbol{c}_2]$, $\boldsymbol{Z}_1 = [\boldsymbol{z}_{1,1}, \boldsymbol{z}_{1,2}]$, $\boldsymbol{Z}_2 = [\boldsymbol{z}_{2,1}, \boldsymbol{z}_{2,2}]$, the singular values of $\boldsymbol{C}$ and $\boldsymbol{Z}_i$ are:

$$\boldsymbol{\sigma}(\boldsymbol{C}) = \sqrt{\frac{||\boldsymbol{c}_1||^2 + ||\boldsymbol{c}_2||^2 \pm ((||\boldsymbol{c}_1||^2 - ||\boldsymbol{c}_2||^2)^2 + 4(\boldsymbol{c}_1^T \boldsymbol{c}_2)^2)^{1/2}}{2}}$$
$$\boldsymbol{\sigma}(\boldsymbol{Z}_i) = \sqrt{1 \pm \boldsymbol{z}_{i,1}^T \boldsymbol{z}_{i,2}} \tag{4}$$

So, $||\boldsymbol{\sigma}(\boldsymbol{C})||_1 = ||\boldsymbol{C}||_*$ is maximized when the centroid vectors have maximal norms (bounded above by 1, since they are the centroids of unit vectors), and are orthogonal to each other. As we saw above the centroid norms is a linear function of within-manifold similarity. Similarly, $||\boldsymbol{\sigma}(\boldsymbol{Z}_i)||_1 = ||\boldsymbol{Z}_i||_*$ is minimized when the within-manifold similarity is maximal. So, both terms in the objective encourage object manifold compression (in the simple case described above the effect is nearly mathematically equivalent). Surprisingly, this implies the first term alone encapsulates both of the key ingredients of a contrastive learning framework, and we do observe that simply maximizing $||\boldsymbol{C}||_*$ is sufficient to learn a useful representation. This is because the compressive role of "positives" in contrastive learning is carried out by forming the centroid vectors, so the objective is

not positive-free even with $\lambda = 0$. For example, if only a single view is used the objective lacks a compressive component and fails to produce a useful representation. We will refer to this version of the objective (with $\lambda = 0$) as "implicit manifold compression." In A.5 we demonstrate empirically that this implicit form effectively reduces $||\boldsymbol{Z_b}||_*$ So, all three factors which determine the manifold capacity (radius, dimensionality, and centroid correlations) can be elegantly expressed in an objective function with a single term, $-||\boldsymbol{C}||_*$.

**Computational Complexity**: The implicit form of our method involves computing a singular value decomposition of $\boldsymbol{C} \in \mathbb{R}^{d \times B}$ which has complexity $\mathcal{O}(Bd \times \min(B, d))$, where $B$ is the batch size and $d$ is the dimensionality of the output. By comparison, many popular methods in SSL involve computing the pairwise cosine similarity between members of the batch, which has complexity $\mathcal{O}(B^2 d)$. Additionally, the complexity of our method (in implicit form) is constant with respect to the number of views used (though the feature extraction phase is linear in the number of views), while pairwise similarity metrics will have quadratic complexity with the number of views.

# 3 METHODS

## 3.1 IMPLEMENTATION DETAILS

**Architecture.** For all experiments we use ResNet-50 (He et al., 2016) as a backbone architecture (for variants trained on CIFAR we removed max pooling layers). Following Chen et al. (2020), we append a small perceptron with one hidden layer to the output of the average pooling layer of the ResNet so that $z_i = g(h(x_i))$, where $h$ is the ResNet and $g$ is the MLP.

**Optimization** We employ a standard set of augmenetations, taking them directly from (Zbontar et al., 2021). For CIFAR and STL-10 we used relatively small batch sizes and the Adam optimizer with a fixed learning rate, for ImageNet-100 we used batch size of 2048 for all three methods and the LARS optimizer with linear warmup and cosine decay. We used a large number of views for small datasets with MMCR (40 for CIFAR and 20 for STL), and a much smaller number of views for ImageNet-100 (we report results for 4 views and 2 views). For more details on specific hyperparameter settings see A.3.

## 3.2 EVALUATION METHODS

**Classification Accuracy.** We follow a standard linear evaluation protocol: the parameters of the encoder, $h$ are frozen and a linear function is trained to classify images using the standard supervised cross entropy loss function, see A.6 for details. We pre-train and evaluate classification performance on CIFAR-10/100, STL-10, and ImageNet-100 (a 100 class subset of the full ImageNet Dataset) (Krizhevsky et al., 2009; Coates et al., 2011; Deng et al., 2009).

**Mean Field Theory Manifold Analysis.** In manifold capacity theory Chung et al. (2018), the radius and dimensionality of manifolds with arbitrary geometry (i.e., not restricted to elliptical form) are determined by the statistics of particular anchor points within the convex hull of the manifold. The anchor points of each manifold are those that uniquely specify the maximum margin separating hyperplanes between said manifold and other manifolds. Though this measure is not well suited for use as an objective, we use it to analyze the capacity of our self-supervised representation (see Fig. 3).

**Subspace Angle.** Besides measuring the size and dimensionality of individual object manifolds we also wish to characterize the degree of overlap between pairs of manifolds. For this, we measure the angle between their subspaces (Knyazev & Argentati, 2002).

**Shared Variance.** Object manifolds will generally have a lower intrinsic dimensionality then the space they are embedded in. Therefore, the data will have low variance along several of the principal vectors used to calculate the set of subspace angles, and so many of the principal angles will have little meaning. To address this limitation we also compute the shared variance between the linear subspaces that contain object manifolds.

**Adversarial Robustness.** Once a linear classifier has been trained on top of the representation, the end-to-end system can be subjected to standard adversarial attacks. We evaluate adversarial robustness under $\ell_\infty$ PGD attacks and AutoAttack (Madry et al., 2018; Croce & Hein, 2020).

Table 1: Top-1 classification accuracies of linear classifiers for representations trained with various datasets and objective functions. Note: for Barlow Twins on ImageNet-100 we report the result from da Costa et al. (2022) which uses a ResNet-18 backbone, as we were unable to obtain better performance. For MMCR on ImageNet-100 we tested both 2 views (matched to baselines) and 4 views, results are formatted (2-view)/(4-view)

| Method | CIFAR-10 | CIFAR-100 | STL-10 | ImageNet-100 |
|---|---|---|---|---|
| Barlow Twins (our repro.) | 90.91 | 67.91 | 89.96 | 80.38* |
| SimCLR (our repro.) | 92.22 | 70.04 | 91.11 | 79.64 |
| Implicit MMCR (ours) | 93.53 | 69.87 | 90.62 | 81.52/82.88 |
| MMCR ($\lambda = 0.01$) | 93.39 | 70.94 | 90.77 | 81.28/82.56 |

**Baseline Models** We chose SimCLR (Chen et al., 2020) and Barlow Twins (Zbontar et al., 2021) for primary comparisons with our model. These methods are recent, simple to implement, and achieve performance comparable to state of the art for self-supervised learning on the Imagenet-1k dataset.

## 3.3 PERFORMANCE

Figure 2 details the evolution of the representation during the course of training. The centroid nuclear norm (Fig. 2b) increases steadily as the centroids become increasingly orthogonal to each other (Fig. 2c) and grow in norm (Fig. 2d). The compression of individual augmentation manifolds is reflected in Fig. 2a. The downstream classification accuracies are reported in Table 1. Subsequent analyses are carried out on the CIFAR-10 dataset using the implicit MMCR model as it achieved the highest performance.

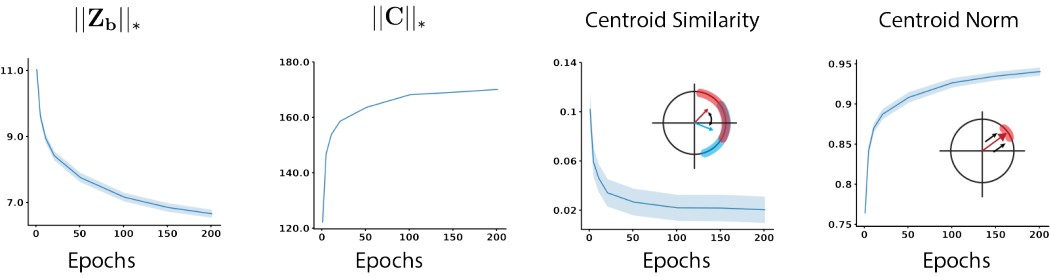

Figure 2: Evolution of various metrics during training. Geometric measures are evaluated on a set of 200 manifolds, each defined by an image drawn from the CIFAR-10 dataset, along with 16 augmentations. Shaded regions indicate a 95% confidence interval around the mean.

## 3.4 REPRESENTATION GEOMETRIC ANALYSIS

In figure 3 we show that our representation, which is optimized using an objective that assumes elliptical manifold geometry, nevertheless yields representations with high mean field manifold capacity (relative to baseline methods). For completeness we also analyzed the geometries of class manifolds, whose points are the representations of different examples from the same class. This analysis provided further evidence that learning to maximize augmentation manifold capacity compresses and separates class manifolds, leading to a useful representation. Interestingly MMCRs seem to use a different strategy than the baseline methods to increase the capacity, namely MMCRs product class/augmentation manifolds with larger radii, but lower dimensionality (Fig. 3)

## 3.5 EMERGENCE OF NEURAL MANIFOLDS VIA GRADIENT COHERENCE

We hypothesize the class separability in MMCRs arises because augmentation manifolds corresponding to examples from the same class are optimally compressed by more similar transformations than those stemming from distinct classes. To investigate this empirically, we evaluate the

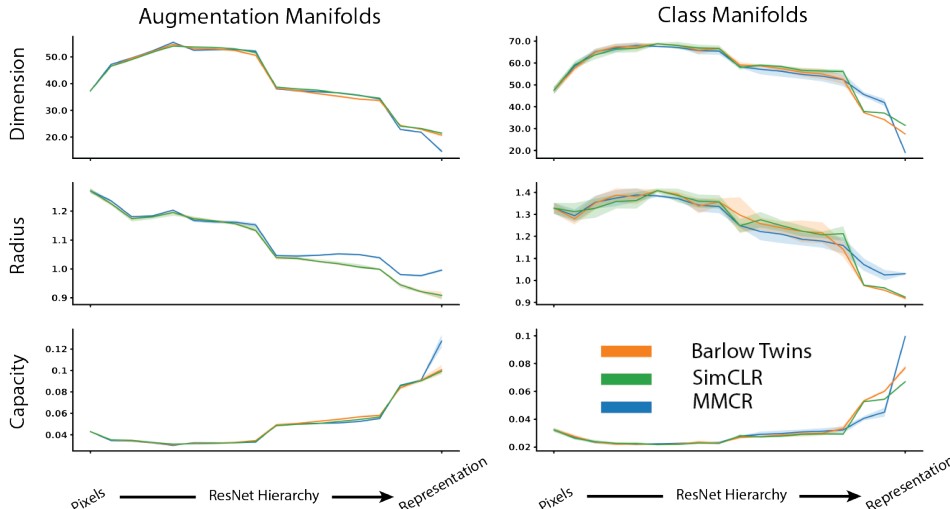

Figure 3: Mean Field Manifold Capacity Analysis. The shared x-axis of all plots is the representational hierarchy, the leftmost entries represent the inputs (pixels) and the rightmost the output of the encoder/learned representation. The top row shows the manifold radius, the middle the dimensionality, and the bottom the resultant capacity. Shaded regions indicate a 95% confidence interval around the mean (analysis was conducted with 5 different random samples from the dataset, see A.4).

gradient of the objective function for inputs belonging to the same class. We can then check whether gradients obtained from (distinct) batches of the same class are more similar to each other than those obtained from different classes, which would suggest that the strategy for compressing augmentation manifolds from the same class are relatively similar to each other. Figure 4 demonstrates that this is the case: within class gradient coherence, as measured by cosine similarity, is consistently higher than across class coherence across both training epochs and model hierarchy.

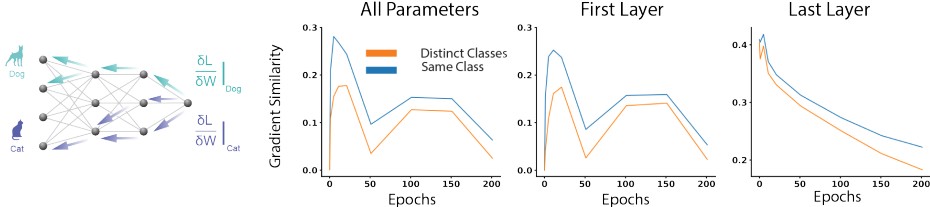

Figure 4: Gradient cosine similarity for pairs of single-class batches. We plot the mean pairwise similarity for pairs of gradients for for different subsets of the model parameters (all parameters, and the first and last linear operators) obtained from single-class-batches coming from the same or distinct classes over the course of training. To the left is a visualization of the fact that single-class gradients flow backward through the model in more similar directions.

### 3.6 MANIFOLD SUBSPACE ALIGNMENT

Within-class gradient coherence constitutes a plausible mechanistic explanation for the emergence of class separability, but it does not explain why members of the same class share similar compression strategies. To begin answering this question we examine the geometric properties of augmentation manifolds in the pixel domain. Here we observe small but measurable differences between the distributions of within-class similarity and across-class similarity, as demonstrated in the top row of figure 5. The subtle difference in the geometric properties of augmentation manifolds in the pixel domain in turn leads to the increased gradient coherence observed above, which over training leads to a representation that rearranges and reshapes augmentation manifolds from the same class in a similar fashion (bottom row of figure 5), thus allowing for the linear separation of classes. Not only

are centroids of same-class-manifolds in more similar regions of the representation space than those coming from distinct classes (Fig 5 third column bottom row) but additionally same-class-manifolds have more similar shapes to each other (Fig 5 bottom row columns 1 and 2).

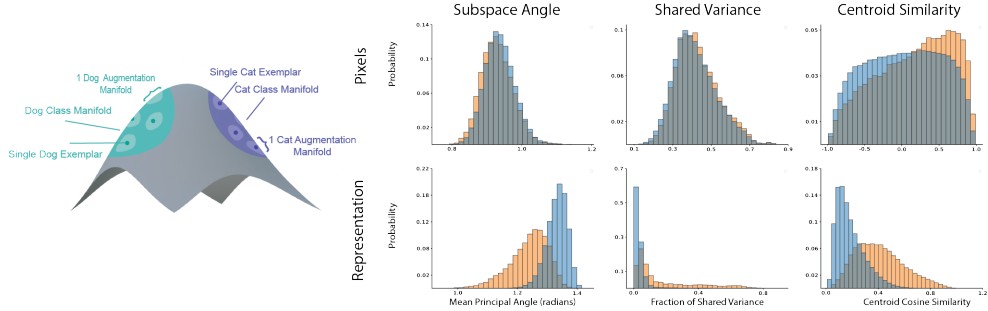

Figure 5: The distributions of various similarity metrics for augmentation manifolds from either the same and distinct classes. In the top row we consider augmentation manifolds in the pixel domain, and in the bottom row we observe how these distributions are transformed by the learned representation. To the left a schematic shows details the exemplar-augmentation manifold-class manifold structure of the learned representation.

We next ask how the representation learned according to the MMCR objective differs from those optimized for other self supervised loss functions. While MMCR encourages centroids to be as close to orthogonal to each other, the InfoNCE loss employed in Chen et al. (2020) benefits when negative pairs are as dissimilar as possible, which is achieved when the two points lie in opposite regions of the same subspace rather than in distinct subspaces. The Barlow Twins (Zbontar et al., 2021) loss is not an explicit function of feature vector similarities, but instead encourages individual features to be correlated across the batch dimension and distinct features to be uncorrelated. In 6 we demonstrate that these intuitions are borne out empirically: the MMCR representation produces augmentation manifold centroids that are significantly less similar to each other than the two baseline methods.

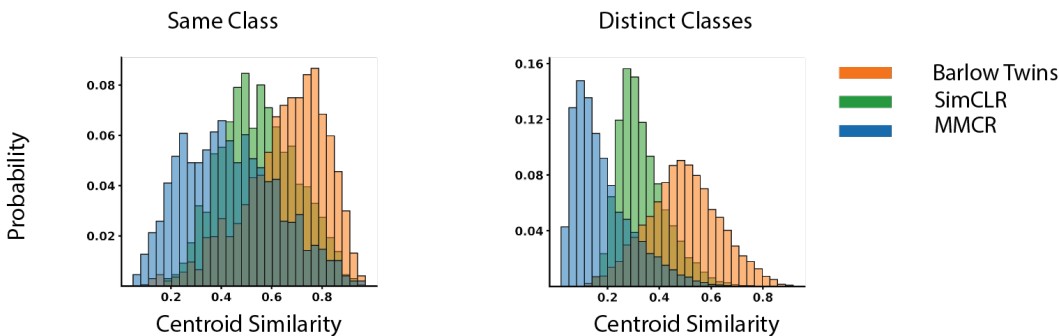

Figure 6: Centroid similarities for models trained according to different SSL objectives. The left panel shows the distribution of centroid cosine similarities for augmentation manifolds for examples of the same class, while the right shows the same distribution for examples from distinct classes.

## 3.7 ADVERSARIAL ROBUSTNESS

Previous works that used similar geometrically motivated loss functions such as the orthogonal low-rank embedding (Lezama et al., 2018) and maximal coding rate reduction Yu et al. (2020) have reported the resulting representations have increased inter-class margins and are more robust to label noise. We therefore tested whether the increased tendency to orthogonalize in MMCR models leads to any benefit in terms of adversarial robustness. In 7 we show that the MMCR model (and attached classifier) is indeed more robust than either Barlow Twins or SimCLR trained models against PGD attacks with a range of strengths (Madry et al., 2018). We found similar results using the stronger

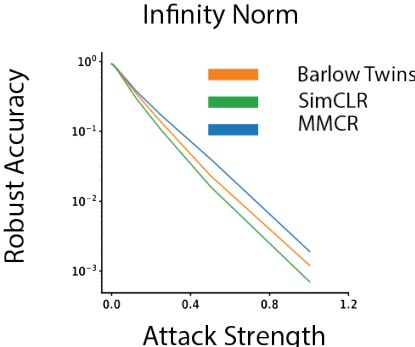

Figure 7: Adversarial Robustness of SSL Models under PGD Attack. For each of the three SSL models with trained classifiers, we apply Projected Gradient Descent (PGD) with $\ell_\infty$-norm perturbation under 50 attack iterations. Inputs were scaled such that their standard deviation was 1.0, so we report the raw attack strengths on the x-axis. Additional details can be found in A.7

AutoAttack protocol (Croce & Hein, 2020), see A.7. Note that Barlow Twins models seem to be more robust than those arising from SimCLR. We speculate that this is a result of the decorrelation (rather than anti-correlation) encouraged by the Barlow Twins objective.

## 4 DISCUSSION

Previous work has proposed manifold capacity theory as a means of understanding the performance of deep neural networks trained using traditional supervised methods (Cohen et al., 2020). Given that manifold capacity theory is particularly well suited for analyzing high dimensional feature spaces, while many SSL methods are motivated by information criterion, we wondered whether manifold capacity could serve as a useful objective function. By approximating the manifold geometries as elliptical, which significantly reduces the computation required to calculate the geometric properties that dictate capacity, we were able to achieve exactly this. Although representational manifold geometries are generally not elliptical, we have demonstrated that this approximation can nonetheless produce a useful learning signal, and leads to networks with high manifold capacity (Fig. 3). Nevertheless, it would be interesting to consider other reductions of the mean field manifold capacity theory that can capture non-elliptical structure of individual manifolds, perhaps by computing higher order statistics of constituent points.

We demonstrate the utility of our approach by applying the method to three small datasets of unlabeled images and demonstrating that the resulting representations can yield similar downstream task performance to two baselines. Furthermore we conducted a gradient based analysis to begin to understand why the self-supervised learning signal is capable of producing useful representations. Finally, motivated by an empirical exploration of the geometrical differences between the representations produced by the three considered methods, we demonstrate that MMCRs can offer improved robustness to adversarial attacks.

Additionally, we leveraged manifold capacity analysis in its full generality to gain insight into the geometry of the MMCR. Intriguingly, our method produces augmentation and class manifolds with lower dimensionality but larger radius than either Barlow Twins or SimCLR (Fig. 3). Future work will seek to understand why this is the case, but more generally this suggests that capacity analysis can be a fruitful way to understand the different encoding strategies encouraged by various SSL paradigms. Another factor that distinguishs MMCRs from other models is a tendency to orthogonalize augmentation manifold centroids and thus form a representation that is globally high dimensional. Given that the recent observations that the representations in visual cortex and high performing models of visual cortex are surprisingly high dimensional (Stringer et al., 2019; Elmoznino & Bonner, 2022), it may be interesting to test how well MMCRs predict neural activity.

In this study, we introduced one specific model of learning using metrics based on a specific theory of representations: self-supervised learning via maximizing neural manifold capacity. More broadly, this work demonstrates the efficacy of a learning objective solely based on representations (rather than relying on classifiers), and motivates new directions of research where representation-level phenomena in neurobiology inform the design choice in neural networks. With recent trends in neuroscience focused on representation geometric observations in neural data, we hope that this work lays foundations for future studies in learning based on representation geometries informed by new discoveries in neuroscience.

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

## A APPENDIX

### A.1 PYTORCH STYLE PSEUDOCODE FOR MMCR

```
# h: encoder
# g: projection head
# B: batch size
# K: number of augmentations
# D: projector output dimensionality
#
# lmbda: trade-off parameter
for x in loader:
    # K randomly augmented views
    x = multi_augment(x) # B x K x H x W

    # push through encoder and projector
    z = g(h(x)) # B x K x D

    # project onto unitsphere
    z = normalize(z, dim=-1)

    # calculate centroids (mean over augmentation axis)
    c = z.mean(dim=1) # B x D

    # calculate singular values
    U_z, S_z, V_z = svd(z) # batch svd
    U_c, S_c, V_c = svd(c)

    # calculate loss
    loss = -1.0 * sum(S_c) + lmbda * sum(S_z) / B

    # backward pass and optimization step
    loss.backward()
    optim.step()
```

### A.2 MEAN FIELD THEORY MANIFOLD CAPACITY BACKGROUND INFORMATION

**Mean Field Theory** Recall the problem setting for manifold capacity analysis: given a set of $P$ manifolds embedded in a feature space of dimensionality $D$, each assigned a random binary class label Chung et al. (2018). Manifold capacity theory is concerned with the question: what is the largest value of $\frac{P}{D}$ such that there exists (with high probability) a hyperplane separating the two

classes? In the thermodynamic limit, where $P, D \to \infty$ but $\frac{P}{D}$ remains finite, the inverse capacity can be written exactly,

$$\alpha_M^{-1} = \mathbb{E}_{\vec{T}}[F(\vec{T})] \tag{5}$$

where, $F(\vec{T}) = \min_{\vec{V}} \left\{ \|\vec{V} - \vec{T}\|^2 \mid g_{\mathcal{S}}(\vec{V}) \geq 0 \right\}$, $\mathcal{S}$ is the set defining the manifold geometry (i.e. the set of vectors $\vec{S}$ that are points on an individual manifold), $\vec{T}$ are random vectors drawn from a white multivariate Gaussian distribution, and $g_{\mathcal{S}}(\vec{V}) = \min_{\vec{S}}\{\vec{V} \cdot \vec{S} \mid \vec{S} \in \mathcal{S}\}$, is the concave support function.

The KKT equations for this convex optimization problem are:

$$\begin{aligned} \vec{V} - \vec{T} - \lambda \tilde{S}(\vec{T}) &= 0 \\ \lambda &\geq 0 \\ g_{\mathcal{S}}(\vec{V}) - \kappa &\geq 0 \\ \lambda \left[ g_{\mathcal{S}}(\vec{V}) - \kappa \right] &= 0. \end{aligned} \tag{6}$$

, where $\tilde{S}(\vec{T})$ is a subgradient of the support function. When the support function is differentiable, the subgradient is unique and equal to the gradient,

$$\tilde{S}(\vec{T}) = \nabla g_{\mathcal{S}}(\vec{V}) = \arg\min_{\vec{S} \in \mathcal{S}} \vec{V} \cdot \vec{S} \tag{7}$$

$\tilde{S}(\vec{T})$ is the unique point in the convex hull of $\mathcal{S}$ that satisfies the first KKT equation, and is called the "anchor point" for $\mathcal{S}$ induced by the random vector $\vec{T}$.

**Equivalent Interpretation of Anchor Points** For a given dichotomy (random binary class labelling) the weight vector of the maximum margin separating hyperplane can be decomposed into a sum of at most $P$ vectors, with each manifold contributing a single vector, which lies within the convex hull of the manifold. The position of said point point is a function of the manifolds position relative to all of the other manifolds in the space and depends on the particular set of random labels. Thus there exists a distribution of separating-hyperplane-determining-points for each individual manifold. Using the "cavity" method it can be shown that these points are none other than the anchor points that are involved in solving the optimization problem described above Gerl & Krey (1994).

**Numerical Solution** To solve the mean field equations numerically, one samples several random Gaussian vectors $\vec{T}$, and then for each $\vec{T}$, $\vec{V}$ and $\vec{S}$ are determined by solving the quadratic programming program given above. The capacity is then estimated as the mean value of $F$ or the samples $\vec{T}$.

**Manifold Geometries** The way the capacity varies in terms of the statistics of the anchor points can be simplified by introducing two key quantities, the manifold radius $R_M$ and manifold dimensionality $R_M$:

$$\begin{aligned} R_M^2 &= \mathbb{E}_{\vec{T}}[\|\tilde{S}(\vec{T})\|^2] \\ D_M &= \mathbb{E}_{\vec{T}}[\vec{T} \cdot \hat{S}(\vec{T})] \end{aligned} \tag{8}$$

where $\hat{S}(\vec{T})$ is a unit-vector in the direction of the anchor point $\tilde{S}$. In particular as discussed in the main text, the manifold capacity can be approximated by $\phi(R_M\sqrt{D_M})$ where $\phi$ is a monotonically decreasing function.

**Elliptical Geometries** In the case where the manifolds exhibit elliptical symmetries, the manifold radius and dimensionality can be written in terms of the eigenvalues of the covariance matrix of the

anchor points:

$$R_M^2 = \sum_i \lambda_i^2$$

$$D_M = \frac{(\sum_i \lambda_i)^2}{\sum_i \lambda_i^2}$$

(9)

So, in this case $R_M$ is the total variability of the anchor points, and $D_M$ is a generalized participation ratio of the anchor point covariance, a well known soft measure of dimensionality.

### A.3 ADDITIONAL PRE-TRAINING INFORMATION

**Settings for CIFAR/STL-10** We take the parameters of each augmentation directly from Zbontar et al. (2021), but for these lower resolution images we omitted Gaussian blurring and solarization augmentations. All models were trained for 500 epochs using the Adam optimizer (Kingma & Ba, 2014) with a learning rate of $1e-3$ and weight decay of $1e-6$. For all three methods we used a one hidden layer MLP with hidden dimension of 512 and output dimension of 128 for the projector head $g$. We swept batch size for each method and chose the one that resulted in the highest downstream task performance. For both SimCLR and Barlow Twins we found that a batch size of 128 was optimal (among 32, 64, 128, 256, and 512) for all 3 datasets. For MMCR there is a trade-off between batch size and the number of augmentations used, and the optimal value of that trade-off is highly dataset dependent. For CIFAR-10 and CIFAR-100 we used batch size of 32 and 40 views, and for STL-10 we used a batch of 64 with 20 views For Barlow Twins we used $\lambda = \frac{1}{128}$ which normalizes for the number of elements in the on-diagonal and off-diagonal terms in the loss. For SimCLR we used the recommended setting of $\tau = 0.5$. The overall performance of both baseline methods (and likely MMCR as well) could be increased with a more thorough hyperparameter search and by employing methodology that more closely matches the original works. For example, both methods would likely benefit from the combination of larger batch size, the use of the LARS optimizer (which is designed for large batch optimization), a learning rate scheduler consisting of linear warm-up followed by cosine annealing, longer training, and the use of more diverse augmentations (i.e. including solarization and gaussian blur). Additionally Barlow Twins reports that the representation can benefit from using a much larger projector network than we use. Because our goal was primarily to demonstrate that MMCR can produce representations that are comparable to these baselines rather than to produce state-of-the-art results on small scale datasets we opted for simplifications wherever possible (using off the shelf Adam for optimization with a fixed learning rate, and fixing architectural hyperparameters like the projector dimensionality).

**Settings for ImageNet-100** For ImageNet we more closely match the pre-training procedures of previous works. We use a batch size of 2048 and a smaller number of views for MMCR (4), and also use the full suite of augmentations from Zbontar et al. (2021). For the sake of efficiency we train for a reduced number of epochs (200). For MMCR and SimCLR we modified the projector hidden dimensionality to be 4096 for the projector head, following the original work Chen et al. (2020). For Barlow Twins we used the recommended 2-layer MLP with hidden and output dimensions of 8192, and set $\lambda = 5e-3$, however these hyperparameters were optimal for the full ImageNet dataset, and not neccesarrily for ImageNet-100. We were unable to achieve better downstream performance using a ResNet-50 backbone than what has previously been reported in the literature for this dataset with a ResNet-18 backbone, therefore we report the ResNet-18 performance reported in (da Costa et al., 2022). For SimCLR we use $\tau = 0.1$ which is the recommended setting for larger batch sizes.

### A.4 DETAILS OF REPRESENTATIONAL ANALYSES

**Manifold Capacity Analysis** For each pre-trained model, we extract layer activations across the ResNet hierarchy after a forward pass of a set of images. For class manifold analysis, the set of images contain 10 classes, where each class has 100 examples. Augmentation manifolds instead have 100 exemplars with 100 examples each. Following Cohen et al. (2020), we take activations from all convolutional layers in ResNet-50 after a ReLU non-linearity. The specific extracted layers highlighted in bold fonts are given by Table 2. The final analysis results are averaged over five data samplings with different random seeds and random projections of intermediate features to lower-dimension spaces (default 5000 dimensions).

Table 2: A Total of 18 Extracted ResNet-50 Layers (in **Bold**) for MFTMA Analysis

| Layer | Type | Conv2d Size (H $\times$ W $\times$ C) |
|---|---|---|
| pixel | **Input** | None |
| conv1 | $\begin{bmatrix} \text{Conv2d} \\ \text{BatchNorm} \\ \textbf{ReLU} \end{bmatrix} \times 1$ | $[7 \times 7 \times 64] \times 1$ |
| conv2_x | $\begin{bmatrix} \begin{bmatrix} \text{Conv2d} \\ \text{BatchNorm} \\ \textbf{ReLU} \end{bmatrix} \times 3 \end{bmatrix} \times 3$ | $\begin{bmatrix} 1 \times 1 \times 64 \\ 3 \times 3 \times 64 \\ 1 \times 1 \times 256 \end{bmatrix} \times 3$ |
| conv3_x | $\begin{bmatrix} \begin{bmatrix} \text{Conv2d} \\ \text{BatchNorm} \\ \textbf{ReLU} \end{bmatrix} \times 3 \end{bmatrix} \times 4$ | $\begin{bmatrix} 1 \times 1 \times 128 \\ 3 \times 3 \times 128 \\ 1 \times 1 \times 512 \end{bmatrix} \times 4$ |
| conv4_x | $\begin{bmatrix} \begin{bmatrix} \text{Conv2d} \\ \text{BatchNorm} \\ \textbf{ReLU} \end{bmatrix} \times 3 \end{bmatrix} \times 6$ | $\begin{bmatrix} 1 \times 1 \times 256 \\ 3 \times 3 \times 256 \\ 1 \times 1 \times 1024 \end{bmatrix} \times 6$ |
| conv5_x | $\begin{bmatrix} \begin{bmatrix} \text{Conv2d} \\ \text{BatchNorm} \\ \textbf{ReLU} \end{bmatrix} \times 3 \end{bmatrix} \times 3$ | $\begin{bmatrix} 1 \times 1 \times 512 \\ 3 \times 3 \times 512 \\ 1 \times 1 \times 2048 \end{bmatrix} \times 3$ |

**Gradient Coherence Analysis** In Fig 4, for each of the classes of CIFAR-10, we generate 100 batches of 32 augmentation manifolds of samples from a specific class (with 40 augmentations each). We then measure the gradient of the loss function for each batch during different stages of training, and compute the cosine similarity between every pair of gradients. Across all stages of training the mean cosine similarity between gradients generated from batches of the same class is larger than those from distinct classes (left column). This observation remains true when isolating the gradients of parameters from different stages of in the resnet-50 hierarchy (center and right columns, respectively).

**Manifold Subspace Alignment** For Fig. 5 we generated 100 samples from the augmentation manifolds of 500 images in the CIFAR-10 dataset. We then measure the mean subspsace angle (left column), fraction of shared variance (middle column) and centroid cosine similarity between each pair of manifolds. The same procedure was used for generating the data for 6.

## A.5 IMPLICIT MMCR EFFECTIVELY REDUCES AUGMENTATION MANIFOLD NUCLEAR NORM

To test whether or not implicit manifold compression actually reduces the mean augmentation manifold nuclear norm, we can vary the value of $\lambda$. Below we see the evolution of bother terms of the loss for several different values of lambda during training on CIFAR-10. For these experiments the batch size was 64 and the number of augmentations per image was 4.0. As shown in 8, the level of compression of individual manifolds is nearly the same across all values of the tradeoff parameter tested.

## A.6 CLASSIFICATION EVALUATION PROCEDURE

During pre-training all models were monitored with a k-nearest neighbor classifier (k=200) and checkpointed every 5 epochs. After pre-training, we trained linear classifiers on all checkpoints whose monitor accuracy was within 1% of the highest observed accuracy, and select the model that

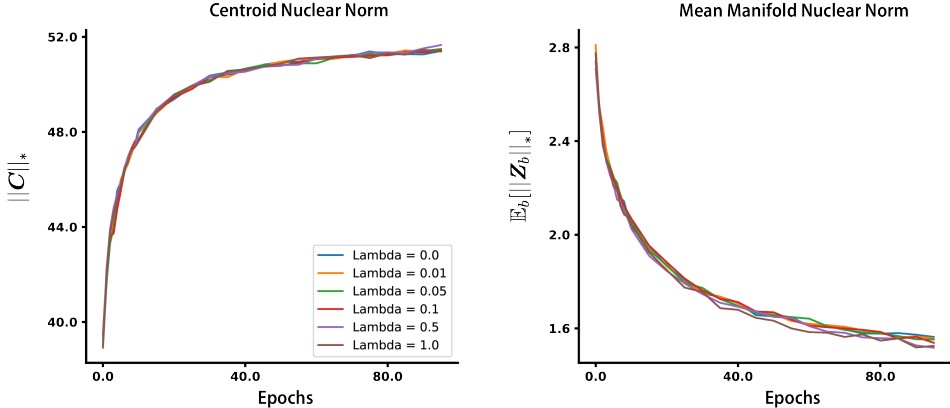

Figure 8: Validation loss values for different values of $\lambda$

achieves the highest linear classification accuracy. Linear classifiers were trained using the Adam optimizer with batch size of 1024 and an initial learning rate of 0.1, which decayed according to a cosine scheduler over the course of 50 epochs. For the linear classifier training, at train time we use the same set of augmentations as during unsupervised pretraining, at test time we only use center cropping and random horizontal flipping.

## A.7 ADDITIONAL DETAILS FOR ADVERSARIAL ROBUSTNESS ANALYSES

In Figure 7, we choose 50 iterations for the PGD $\ell_\infty$-norm since it guarantees a robust accuracy value not far away from asymptotically larger PGD attack iterations (Madry et al., 2018; Croce & Hein, 2020). In our experiment, we have shown that the PGD attack indeed converges in a similar fashion (See Figure 9). However, the robust accuracy for MMCR tends to converge at larger PGD attack iterations.

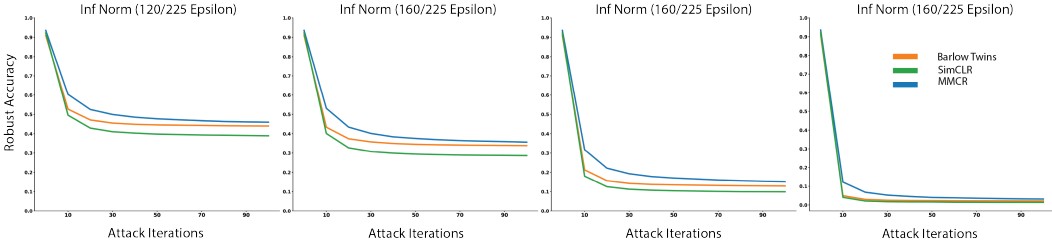

Figure 9: Convergence for different settings of adversarial attack strengths

We therefore also analyzed the robust accuracies for the three SSL methods with varying iterations across all epsilon attack strength. Figure 10 shows MMCR exhibits a significantly higher robust accuracy compared to Barlow-Twins and SimCLR in the low iterations regime.

Aside from the standard PGD adversarial attack, we also tested three SSL methods under the AutoAttack protocol. The $\ell_\infty$-norm AutoAttack accuracy is given by Table 3.

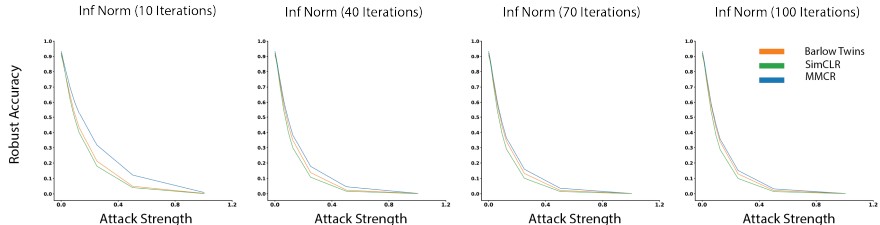

Figure 10: PGD $\ell_\infty$-norm attack with varying iterations.

Table 3: AutoAttack $\ell_\infty$-norm Robust Accuracy

| Method | Clean Accuracy | Eps = 40/255 | Eps = 160/255 |
|---|---|---|---|
| Barlow Twins (our repro.) | 90.91 | 74.55 | 31.53 |
| SimCLR (our repro.) | 92.22 | 72.48 | 26.37 |
| Implicit MMCR (ours) | 93.53 | **75.88** | **32.47** |

