# OpenReview forum: "Self-Supervised Learning of Maximum Manifold Capacity Representations"
_ICLR.cc/2023/Conference — Submitted to ICLR 2023_

### Official Review · Reviewer_kyPr · 2022-10-24

**Confidence:** 4
**Correctness:** 3
**Technical Novelty And Significance:** 2
**Empirical Novelty And Significance:** 3
**Recommendation:** 5

**Clarity, Quality, Novelty And Reproducibility:**

The paper is relative clear but the presentation could be improved. The perspective of interpreting the nuclear norm from a manifold capacity is new to contrastive self-supervised learning. But emplying the nuclear norm as a loss function to contrastive learning is not novel.

**Strength And Weaknesses:**

The strengths of the paper:

+ It is interesting and somewhat novel to introduce the measure of manifold capacity into contrastive self-supervised representation learning.


The weaknesses of the paper:

- The novelty of introducing the manifold capacity into contrastive learning is somewhat weak. In this paper, the manifold capacity concept at the end of the day again turns out to be optimizing with a loss function of the nuclear norm on the centroids.  Nevertheless, the similar idea of using nuclear norm into contrastive learning has been explored in (Wang et al. TPAMI 2022). The differences from (Wang et al. TPAMI 2022) is not clearly discussed.

- It is correct to relate the mean to the average consine similarity as Eq. (3). However, it seems problematic to interprete the rationale to keep only the first term in Eq. (2) by introducing a norm inequality, i.e., the nuclear norm is bounded from below by the Frobenius norm. Note that the optimization is to maximize the nuclear norm term, it might not guarantee to maximize its lower bound.

Moreover, the reviewer is confused by the mixed expressions in Eq. (4).

- In experiments, the computation time is not mentioned. Compared to the existing method, is there some advantage in computation time?

**Summary Of The Paper:**

The paper presents a contrastive learning approach by maximizing manifold capacity via a nuclear norm for self-supervised representation learning. Experiments show that the proposed approach is able to yield better linear evaluation performance, extract sementically relevant features, and be more robust to adversarial attack.

**Summary Of The Review:**

While introducing the nuclear norm from the perspective of measuring manifold capacity is new to contrastive learning, overall the novelty is marginal due to that the nuclear norm as a loss function has been used in contrastive learning.  The empirical evaluation is okay but not that strong.

---

> ### Author Response · Authors · 2022-11-15
> **Response to Reviewer kyPR**
>
> Thank you for your helpful review, our point-by-point responses are below.
>
> ### Relationship to Wang et. al 2022 ###
> The work of Wang et al. is certainly related to ours, and we agree it is worth making more clear how our contribution differs from theirs. The low rank promoting prior method (LORAC) explores how a term like $||\boldsymbol{Z_b}||_*$ in our notation can be incorporated into existing SSL frameworks (i.e. Momentum contrast in their case) as a prior or regularizer. Our work is a more radical departure from traditional SSL in that we not only impose a low-rank constraint on the representation of an ensemble of positive samples, we demonstrate that learning to be globally high rank is effective for representation learning. I.e. while LORAC combines a low rank prior with an InfoNCE based likelihood, our work makes the more symmetric choice of pairing a low-rank prior with a high-rank likelihood. Furthermore, we demonstrate that when combined with the common scheme of normalization and less common scheme of forming centroid vectors, it is not actually necessary to compute  for each image in the batch (see below). We have expanded the discussion of the relationship between our work and Wang et al. to more clearly show the differences.
>
> ### Effectiveness of the Implicit Form of the Objective ###
> We agree that there is no guarantee that optimizing that optimizing $||\boldsymbol{C}||_*$ will in turn reduce $||\boldsymbol{Z_b}||$, though one term bounds the other. However we do empirically observe this to be the case. We conducted additional experiments on CIFAR-10 where we used $\lambda =$ [0.0, 0.01, 0.05, 0.1, 0.5, 1.0], and observed little to no difference in the values of $||\boldsymbol{Z_b}||$ that were achieved. We have added these experiments to the appendix of the manuscript (see section A.4)
>
> ### Computational Complexity ###
> Thank you for pointing out this possibility. See our general comment for a discussion of the computational complexity of our method in comparison to existing methods in the same setting. We have added a similar discussion to the manuscript as well.
>
> ### Empirical Evaluation ###
> We have added evaluation on ImageNet-100, which is a significantly larger dataset than those considered originally. See our general response and updated manuscript for more details.

---

> > ### Author Response · Authors · 2022-12-06
> > **Response Update**
> >
> > We have now conducted additional experiments that significantly strengthen the empirical validation of our work (results for the standard ImageNet-1k dataset, see our update to the general reply above). Because we were unable to finish these experiments during the initial discussion period we wanted to be sure to bring them to your attention now. We would include these experiments in the camera-ready manuscript if accepted.
> >
> > We have also done our best to address the issues raised in your initial review in the revised manuscript, please let us know if there is anything else we could do to improve the paper!

---

### Official Review · Reviewer_Xa59 · 2022-10-24

**Confidence:** 4
**Correctness:** 3
**Technical Novelty And Significance:** 2
**Empirical Novelty And Significance:** 2
**Recommendation:** 5

**Clarity, Quality, Novelty And Reproducibility:**

**Clarity** the biggest problem concerning clarity is that the paper is not self-contained and the reader needs to read the two previous papers on manifold capacity to truly understand the intuition behind using such a loss. The rest of the paper is clear and well written.

**Novelty** the novelty and contributions seem minor:
     - methodology: the paper seems to essentially use a metric that had previously been proposed for evaluating representations and optimizing it
    - theory: there is no new theory
    - experiments: empirical gains are minor and only on small datasets


**Reproducibility**: code and hyperparameters are provided

**Strength And Weaknesses:**

**Update**: I've updated my score since he authors added experiments on a non toy ImageNet-100 dataset, but I still find the contributions of the work minor.

**Strengths**
- **Shedding light on recent theory on manifold capacity** this paper sheds light on recent advancements on manifold capacity that can be of interest to the representation learning community.  These findings are the starting point of the paper (not a contribution) but disseminating these tools to the ML community has its own value given that those works were not published in ML venues. Unfortunately, the paper only provides some high-level informal discussion about those tools.
- **problem is of interest to the community** representation learning is definitely of interest (even more so if it is easier to analyse theoretically**
- **well written** the paper is generally well written and clear (besides the fact that it is not self contained as discussed below)

**Weaknesses**
- **Not self-contained** I had to partially read Chung 2018 and Cohen 2020 to understand the really understand section 2 (what is manifold capacity and why one could care). From the current paper, it is not even clear which definition they use for important terms that are used throughout (eg manifold, manifold radius, manifold effective dimensionality, and capacity).
- **Unconvincing/small experiments** the method performs similarly to standard SSL on small datasets, it is unclear what to make out of that. Why is that important?
- **Unclear or minor contributions**
    - methodology: the paper essentially pretrains using a loss that was previously proposed for evaluation
    - theory: no theory, not even proper discussion of the theory that inspired them
    - experiments: the method performs essentially as well as standard SSL on small datasets. It is unclear what are the benefits of the method.
- **Lack of discussion of previous work** there are many previous works that provides a similar intuition. Eg the following papers and reference therein:
    - [[1]](https://arxiv.org/abs/2209.06235) this paper characterizes the optimal geometry of representations for maximizing linear probing capacity and generalization (and investigates them in practice). One possible such optimal representation is the sETF, which I believe is also what you are suggesting (ie minima to your loss.
    - neural collapse literature: there's an entire line of work that analysis the geometry of representations and in particular sETF. Although this is mostly about supervised learning I think it is very related, one example is [[2]](https://arxiv.org/abs/2102.08817) which analysis contrastive learning
   - [[3]](https://arxiv.org/abs/2106.04156) their loss (eq 6) is pretty related to your loss although their derivation comes from the decomposition of the augmentation graph.


**Summary Of The Paper:**

This is an empirical paper that proposes a new self-supervised learning objective, which consists in minimizing of nuclear norm of the representation of positive examples (~low rank => low variability) while maximizing that of negative examples (more precisely their centroids). This is inspired by recent work on manifold capacity, ie, the number of linear classifiers that can separate point clouds/manifolds. The authors show on small datasets (CIFAR10/100 and STL10) that such objective performs well, and analyze how the geometry (eg distance between positives and negatives) changes during training and across layers.

**Summary Of The Review:**

I enjoyed reviewing the paper in that it brought to my attention novel work on manifold capacity that was developed outside of ML. Unfortunately, the paper is not self-contained, and provides no theory, and no convincing experiments. Although I do not think that the current version should be accepted, I think that future versions will be of interest to the community. If the main contribution is shedding light on the theory of manifold capacity (which would be fine), the authors should discuss those tools more formally (definitions, main statements). If the contributions go beyond that then the authors should scale up their experiments or provide some theoretical guarantees using the manifold capacity theory.

I am happy to increase my score if actions take action on those suggestions or provide convincing arguments as to why those are not necessary.

---

> ### Author Response · Authors · 2022-11-15
> **Response to Reviewer Xa59**
>
> Thank you for your careful reading of our paper, we feel the manuscript has been improved by addressing several of the limitations raised. Below are our point-by-point responses.
>
> ### Self Containedness ###
> Our goal is to introduce the concept of manifold capacity to the ML community by demonstrating how contrastive SSL can be viewed as a capacity maximization problem. This required a significant simplification of the general theory to make the relevant metrics feasible to use as an objective function. To better communicate this, we’ve amended the main text at the beginning of section 2.1. Additionally we will be adding an Appendix section that gives significantly more detail on the mean field theory in its full generality. Although we do not feel that the papers detailing the general theory are required reading for understanding either the contribution or framing of this work, we do hope that some readers will have their interest piqued enough to read some of the early work on the topic!
>
> ### Empirical Evaluations ###
> We have conducted, and will include, additional experiments using ImageNet-100 (see general responses, above).
>
> ### Connections to Previous Work ###
> [1, 2] sETFs and Neural Collapse: Thank you for bringing these recent interesting results to our attention. After an initial reading, it does seem to us that a centroid sETF would be an optimal solution to our objective function. Specifically, our objective can be interpreted as learning an sETF by directly optimizing singular values, rather than via a logistic regression-like loss as is more common in SSL. We include a discussion of this connection in the updated related works section of the manuscript.
>
> [3] Spectral Contrastive Learning: Although there is some similarity between our loss function and the spectral contrastive loss, it is more similar to SimCLR (as the authors of that paper point out) differing mostly in how nonlinearities are applied to the feature-by-feature similarity matrix. The key difference between our metric and spectral contrastive loss is that we do not rely on pairwise comparisons.

---

> > ### Comment · Reviewer_Xa59 · 2022-11-17
> > **Thank you for the update I will raise my score but still find the contributions minor.**
> >
> > Thank you for your answer, I've read your general rebuttal and also the updated manuscript.
> >
> > The updated manuscript is somewhat better in terms of discussion about related work and discussion contributions (eg computational issues with manifold of arbitrary geometry). Having ImageNet-100 is also much better, it is still small but at least it is a standard SSL dataset for empirical work under computational constraints.  I nevertheless still find the contributions relatively minor, and the (even simplified) discussion / summary of manifold capacity insufficient given that the main contribution seems to be introducing this to the SSL community. Given that I do think that others (like me) might be interested in the use of the manifold capacity theory for SSL and that you at least have non toy experiments, I will update my score.
> >
> > Concerning ImageNet-100 results my only worry about those results is that, if I understand correctly, you only show MMCR results with setting even though the baselines use only 2 views. Please add the results of MMCR with 2 views (even if they are worst than baselines, you can always explain why) or add baselines with 4 views (eg SwAV, which introduced the multiview setting). Note that if the issue is computational complexity of other multiview methods, SwAV shows how to get multiview gains without increasing much the computational complexity (see 2×160 + 4×96 in section 4.3).

---

> > > ### Author Response · Authors · 2022-11-17
> > > **Thank you for your response.**
> > >
> > > Thank you for taking the time to read the revised submission. We have now added an appendix to the manuscript that gives a more detailed description of the key results of the general theory to bolster the value of this work as an introduction to manifold capacity geared towards the SSL community.
> > >
> > > Regarding ImageNet-100 baselines, we are currently running 2-view MMCR and will report the results as soon as possible.

---

> > > > ### Author Response · Authors · 2022-11-18
> > > > **2-View Baseline Results**
> > > >
> > > > We have updated our general response and manuscript to include results for MMCR using 2-views on ImageNet-100. We observed an approximately 1.3% decrease in the top-1 accuracy of downstream classification accuracy compared to the 4-view version.
> > > >
> > > > We assume the reason for this decrease in performance is simply that fewer samples are being used to estimate each augmentation manifold centroid, resulting in a noisier learning signal.

---

> > > > > ### Comment · Reviewer_Xa59 · 2022-11-20
> > > > > **Thanks!**
> > > > >
> > > > > Thank you for running the additional experiment. I agree with your hypothesis. In previous work using multiple views  also increased performance by 1-3 percentage  points, so that is not very surprising!

---

> > > > > > ### Author Response · Authors · 2022-12-06
> > > > > > **ImageNet-1k Results**
> > > > > >
> > > > > > We have now conducted experiments on the full ImageNet-1k dataset (see our updated general response above). Implicit MMCR with 4 views is competitive with SOTA SSL methods when pre-training for 100 epochs. We would like to thank you for pushing us to scale up the experiments as we agree these results make the work much more compelling.

---

### Official Review · Reviewer_HB7z · 2022-10-25

**Confidence:** 3
**Correctness:** 3
**Technical Novelty And Significance:** 2
**Empirical Novelty And Significance:** 2
**Recommendation:** 6

**Clarity, Quality, Novelty And Reproducibility:**

This work can be considered as an interesting extension of previous manifold capacity work to self-supervised learning and provide us a different way to think about SSL outside the typical information theoretic direction.

Also good to see author provides the code and data in the supplementary material.

**Strength And Weaknesses:**

Strengths:

Introduction the concept of manifold capacity to SSL is quite interesting and seems to be the first work in this direction (unless I missed some prior works). By utilizing the nuclear norm with solid theoretical support, this proposed loss function in eq. 2 is well presented as the combination of "encourage separability" plus "manifold compression".

This "implicit manifold compression" version of MMCR with only the first term , is also encourage object manifold compression and can learn useful representation. Also, the connection to previous work MCR2 is discussed, good to see the similarities and difference from this line of work (probably can have more).

Both Figure 1 and the analysis in Section 2.2 are good to have, to help understand intuition behind the proposed objective function.

Experimental results on image data across a number of tasks from classification accuracy to manifold structure related analysis, this is helpful to support the contributions.

Weaknesses:

Synthetic data is often helpful to give us insights, and for this paper seems there is no results from simulated data experiments. It should be good to see how exactly "manifold capacity" been numerically optimized under simple toy examples.

Given all experiments are performed under ResNet-50 as the backbone architecture, should be nice to have some computational complexity discussion for the proposed MMCR method, as this is less clear overall.

**Summary Of The Paper:**

Maximum Manifold Capacity Representation (MMCR) is proposed as a novel self-supervised learning framework by maximize the number of linearly separable object manifolds, which is interesting to see, and different from many SSL methods are inspired by information & entropy criterions.

Experimental results on several computer vision data (e.g., CIFAR-10, STL-100, CIFAR-100) are included for a number of applications, e.g., classification accuracy, manifold capacity analysis, subspace alignment, adversarial robustness, etc.

**Summary Of The Review:**

This is an interesting work in the SSL domain, the introducing of the concept of manifold capacity is informative, and overall technical contribution seems to be marginal but solid. Perhaps the empirical result is relative limited as no good results from large data set included yet, and my recommendation for this paper is 6 "marginally above the acceptance threshold".

---

> ### Author Response · Authors · 2022-11-15
> **Response to Reviewer HB7z**
>
> Thank you for your positive review of our paper. Find below our responses to the specific weaknesses mentioned
>
> ### Synthetic Datasets ###
> We agree that applying our objective to a toy dataset could help to build intuition. We have begun designing an experiment that involves optimizing the capacity metric for inputs drawn from a mixture of Gaussian distribution using a simple network, but do not have any results to share yet.
>
> ### Computational Complexity ###
> We have added a discussion of computational complexity to the manuscript (see details in our general response above), this was indeed an oversight in the original submission.
>
> ### Empirical Evaluation ###
> We have added experiments on the significantly larger ImageNet-100 data set, where our method achieves strong performance (see general response).

---

### Official Review · Reviewer_Syek · 2022-10-25

**Confidence:** 3
**Correctness:** 3
**Technical Novelty And Significance:** 2
**Empirical Novelty And Significance:** 3
**Recommendation:** 5

**Clarity, Quality, Novelty And Reproducibility:**

The paper is well-written. The quality is high, but the novelty of the paper is questionable. It applies the existing manifold capacity to contrastive learning. However, the superiority analysis is not clear. Please see detailed comments above.

**Details Of Ethics Concerns:**

I have no ethics concerns for this paper.

**Strength And Weaknesses:**

Strength:
Many self-supervised learning methods try to optimize an approximation of the mutual information between representations of different views. The paper uses the manifold capacity to maximize the number of linearly separable object manifolds. It is a new approach that prevents the collapse in the representation space in contrastive learning.

Weakness:
1.	It is better to discuss the relation with dimensional collapse in contrastive learning. Also, it does not compare with the methods addressed the limitation. For example, in the following paper, it uses the subspaces to represent global image manifold and develops new method to prevent the collapse in the representation space.
Li Jing, Pascal Vincent, Yann LeCun, Yuandong Tian. Understanding Dimensional Collapse in Contrastive Self-supervised Learning. ICLR 2022.
2.	It is better to detail the solution process of (2). For large image sets, how to solve problem (2) efficiently?
3.	It is better to discuss the possibility to use sophisticated manifold geometries instead of elliptical.

**Summary Of The Paper:**

The paper introduces the self-supervised learning via maximizing manifold capacity. Specifically, it incorporates the manifold capacity into a contrastive objective. It maximizes the extent of the global image manifold and minimizes every object manifold. The paper approximates the manifold geometries as elliptical to reduces the computation. Experiments on object recognition and robustness to adversarial attacks test the effectiveness of the proposed approach.

**Summary Of The Review:**

Overall, this paper could be an interesting algorithmic contribution. However, my main concern is the novelty of the paper. Currently, I am leaning toward rejecting the paper.

---

> ### Author Response · Authors · 2022-11-15
> **Response to Reviewer Syek**
>
> ### Relationship to Jing et al. 2022 ###
> Thank you for your thoughtful comments. We agree that it is useful to view our method as a novel method for preventing dimensional collapse in self-supervised learning. Specifically, Jing et al. detail the mechanisms by which optimizing the InfoNCE loss using the standard SSL paradigm leads to dimensional collapse by examining the dynamics of singular values during gradient descent. Their analysis sheds light on the need for projector networks to prevent dimensional collapse, and they demonstrate that using a fixed (rather than learnable) low dimensional projection is sufficient.
>
> Our method can indeed be understood as an alternative method for collapse prevention as the global term $||\boldsymbol{C}||_*$ is a soft measure of the rank of the representation space and is directly maximized. It would be interesting to explicitly measure the distribution of singular values in networks trained according to our loss to compare the two approaches, or to investigate whether in light of this difference our method requires the use of a projector at all. We have updated the related works section of the manuscript to include a discussion of this connection.
>
> ### Novelty ###
> While we are in principle optimizing a metric that has been derived previously, our work outlines the appropriate modifications and reductions that are necessary to use said principle as an objective function. Please see our general response for a more nuanced description. We have also updated Section 2.1 of the manuscript to better reflect how our contribution builds on the previous theoretical foundation.
>
> ### Efficiency ###
> We have added a section detailing the computational complexity of calculating the loss while optimizing our objective function (in the implicit case). We have also added results for a significantly larger dataset (ImageNet-100) demonstrating the objective can be used for larger scale problems than those considered in the initial submission.
>
> ### Other Geometries ###
> While optimizing for capacity in the case of arbitrary geometry is computationally unfeasible, we agree it would be interesting to consider other non-elliptical geometries. Perhaps an interesting extension of this work could achieve this by calculating higher order statistics of points on each manifold. We have added a note in the Discussion alluding to this possibility.

---

> > ### Author Response · Authors · 2022-12-06
> > **Response Update**
> >
> > We have now conducted additional experiments that significantly strengthen the empirical validation of our work (results for both ImageNet-100 and the standard ImageNet-1k dataset, see our update to the general reply above). Our method is competitive with state of the art in SSL when pretraining for 100 epochs. Given we were unable to complete these experiments during the initial review period we wanted to be sure to bring these results to your attention here.
> >
> > We have also revised the manuscript to include a discussion of dimensional collapse as you suggested in your initial review, please let us know if you have any other suggestions that would help improve the paper.

---

### Author Response · Authors · 2022-11-15
**General Response**

We thank the reviewers for their careful consideration of our submission, which has helped us to improve the work.  We begin by addressing several points that were raised by multiple reviewers:


### Contribution Novelty ###
Multiple reviewers correctly point out that the crux of our contribution is to demonstrate that an existing metric for manifold capacity can be used as an optimization objective. We believe this is a meaningful and nontrivial contribution: it was not at all obvious that this metric could serve as a sole objective (i.e., without additional regularization) for self-supervised learning, especially since the original metric was far too computationally costly to be used for this purpose.  We developed an ellipsoidal approximation (that is roughly 500x faster to evaluate than the full metric, when tested on 100 manifolds with 100 samples each), demonstrated its effectiveness in training on small to medium-sized datasets, and analyzed its consistency and robustness to adversarial attack.

### On the Computational Complexity ###
A primary advantage of our method over alternatives is that the singular value decomposition is computed on the augmentation centroid matrices rather than operating on the feature-by-feature covariances, or batch-by-batch similarity matrices. Because the centroid matrix has shape $B \times D$ where $B$ is the batch size and $D$ is the dimensionality of the output, the complexity of the SVD is $\mathcal{O}(BD*min(B, D))$. This complexity remains fixed in the multiview setting, since the centroid matrix shape is independent of the number of views. The SimCLR objective function relies on pairwise comparisons, with a cost in the multiview setting  of $\mathcal{O}(B^2N^2D)$  (here N is the number of views). While we do not know of any multiview implementation of Barlow Twins, if one were to compute all possible cross correlation matrices the complexity would be $\mathcal{O}(N^2BD^2)$ as $\mathcal{O}(BD^2)$ is the complexity for computing one of the cross correlation matrices.

### Application to Larger Scale Datasets ###
Multiple reviewers pointed out that the potential impact of our work hinges on whether or not it can be scaled to larger datasets. We have now completed a new set of experiments on the ImageNet-100 dataset, and find that our method achieves (to the best of our knowledge) SoTA accuracy (these results are taken from https://arxiv.org/pdf/2209.07999.pdf with the exception of SimCLR and our method, see the updated manuscript for more details):


| SimSiam  | BYOL | CorInfoMax | SimCLR | Barlow Twins | Implicit MMCR (4 views)| Implicit MMCR (2 views) |
| ----------- | ------- | ------------- | --------- | --------------- | ---------------------------- | --------------------------- |
| 81.6         | 78.76  | 82.64           | 79.64    | 80.38*             | 82.88                               |  81.52                            |

*Barlow Twins result uses a ResNet-18 rather than a ResNet-50 backbone

### Revision Summary ###
We have made the following changes to the manuscript based on reviewer feedback:
- Revised the introduction of the general manifold capacity theory to make our contribution more clear. We have also added an appendix that gives a more detailed summary of the original theory of manifold capacity.
- We conducted additional experiments to validate that the implicit version of our objective functions in much the same way as the explicit form.
- We conducted additional experiments on the significantly larger ImageNet-100 dataset, and found that our method compares favorably to the considered baselines in this setting as well.
- We added a discussion of computational complexity to the manuscript.
- We revised the Related Works section of the manuscript to better connect with existing works as suggested by reviewers.

---

> ### Author Response · Authors · 2022-12-06
> **General Response Update (ImageNet-1K results)**
>
> We have now completed additional experiments evaluating our method on the ImageNet-1K dataset. We pre-train for 100 epochs and then perform standard linear evaluation (results for other methods in this setting are taken from https://arxiv.org/pdf/2209.07999.pdf):
>
>
> | SimSiam  | BYOL | CorInfoMax | SimCLR | Barlow Twins | Implicit MMCR (2 views)| Implicit MMCR (4 views) |
> | ----------- | ------- | ------------- | --------- | ---------------  | ----------------------------| --------------------------- |
> | 68.1         | 69.3    | 69.08           | 66.5      | 68.7                | 67.25                              |  69.62                             |
>
> This result shows that MMCR is not limited to small or medium scale problems, and is indeed competitive with state of the art methods.

---

### Decision · Program_Chairs · 2023-01-20

**Decision:**

Reject

**Justification For Why Not Higher Score:**

This work provides an interesting idea to suppress the risk of mode collapse for self-supervised learning. However, the experimental part is too weak because of the lack of strong baselines. Although the authors added more baselines like BYOL and SimSiam in the rebuttal phase, some existing strategies used to mitigate mode collapse [a, b, c] are still not considered. Without the comparisons of these methods on performance and complexity, the value of this work is not very convincing.

[a] Hua, Tianyu, et al. "On feature decorrelation in self-supervised learning." Proceedings of the IEEE/CVF International Conference on Computer Vision. 2021.

[b] Jing, Li, et al. "Understanding dimensional collapse in contrastive self-supervised learning." arXiv preprint arXiv:2110.09348 (2021).

[c] Bardes, Adrien, Jean Ponce, and Yann LeCun. "Vicreg: Variance-invariance-covariance regularization for self-supervised learning." arXiv preprint arXiv:2105.04906 (2021).


**Justification For Why Not Lower Score:**

N/A

**Metareview: Summary, Strengths And Weaknesses:**

In this submission, the authors proposed a new self-supervised learning framework to suppress the risk of mode collapse when learning data representations. The proposed method is a new member of contrastive learning, motivated by an evaluation measurement called manifold capacity for the quality of data representation. In particular, by maximizing the number of linearly separable object manifolds, the proposed method ensures the learned representations with sufficient distinguish power and diversity.

Strengths:
(1) The idea is clear and easy to follow. The manifold capacity maximization framework is reasonable in my opinion.

Weaknesses:
(1) All the reviewers agree that the experimental part is too weak. Although the authors introduced more baselines in the rebuttal phase, existing solutions to dimensional collapse or mode collapse (e.g., the references shown below) are not considered.